

# Structural and functional analysis of the roles of the HCV 5′ NCR miR122-dependent long-range association and SLVI in genome translation and replication

Kirsten Bentley[1], Jonathan P. Cook[2], Andrew K. Tuplin[3] and David J. Evans[1]

[1] BSRC and School of Biology, University of St Andrews, St Andrews, UK
[2] School of Life Sciences, University of Warwick, Coventry, UK
[3] The Faculty of Biological Sciences, University of Leeds, Leeds, UK

Corresponding author
Kirsten Bentley,
kb209@st-andrews.ac.uk

## ABSTRACT

The hepatitis C virus RNA genome possesses a variety of conserved structural elements, in both coding and non-coding regions, that are important for viral replication. These elements are known or predicted to modulate key life cycle events, such as translation and genome replication, some involving conformational changes induced by long-range RNA–RNA interactions. One such element is SLVI, a stem-loop (SL) structure located towards the 5′ end of the core protein-coding region. This element forms an alternative RNA–RNA interaction with complementary sequences in the 5′ untranslated regions that are independently involved in the binding of the cellular microRNA 122 (miR122). The switch between 'open' and 'closed' structures involving SLVI has previously been proposed to modulate translation, with lower translation efficiency associated with the 'closed' conformation. In the current study, we have used selective 2′-hydroxyl acylation analysed by primer extension to validate this RNA–RNA interaction in the absence and presence of miR122. We show that the long-range association (LRA) only forms in the absence of miR122, or otherwise requires the blocking of miR122 binding combined with substantial disruption of SLVI. Using site-directed mutations introduced to promote open or closed conformations of the LRA we demonstrate no correlation between the conformation and the translation phenotype. In addition, we observed no influence on virus replication compared to unmodified genomes. The presence of SLVI is well-documented to suppress translation, but these studies demonstrate that this is not due to its contribution to the LRA. We conclude that, although there are roles for SLVI in translation, the LRA is not a riboswitch regulating the translation and replication phenotypes of the virus.

Subjects Microbiology, Molecular Biology, Virology
Keywords Hepatitis C virus, RNA–RNA interaction, SHAPE, miR122, Riboswitch

## INTRODUCTION

Hepatitis C virus (HCV) belongs to the genus *Hepacivirus* in the family *Flaviviridae* and, despite the recent development of novel and effective therapies (*Gao et al., 2010*;

*Lawitz et al., 2013*; *Welzel et al., 2017*), infects approximately 185 million people globally, causing significant levels of chronic liver disease and hepatocellular carcinoma (*Mohd Hanafiah et al., 2013*). Like other flaviviruses HCV possesses a single-stranded, positive(mRNA)-sense genome packaged into an enveloped virus particle (*Chambers et al., 1990*). The virus genome expresses a single extensive open reading frame, flanked by highly structured 5′ and 3′ untranslated regions (UTRs) which, upon delivery to the cytoplasm, is translated to yield a single polyprotein. The latter is co- and post-translationally processed to generate the proteins required for genome replication and particle formation. Thereafter, in a relatively poorly-understood process the genome must act as the template for both translation of the polyprotein and replication resulting in new progeny genomes and, eventually, virus particles. The two processes of translation and replication, unless temporally separated or compartmentalised, must be mutually exclusive and are therefore likely to be controlled.

The limited coding capacity of small RNA viruses necessitates the genome being multi-functional, with control of key events in the replication cycle being influenced by the presence of RNA secondary structures involved in either RNA–RNA interactions and/or the recruitment of viral or host factors (reviewed in; *Li & Nagy, 2011*; *Nicholson & White, 2014*; *Tuplin, 2015*). For HCV, a number of RNA structures and RNA interactions have been identified throughout the genome and linked to a variety of functional roles (reviewed in; *Adams, Pirakitikulr & Pyle, 2017*; *Niepmann et al., 2018*). Known or predicted conformational changes induced by long-range RNA–RNA interactions in the HCV genome have been described as molecular switches, potentially modulating translation by facilitating the switch from protein synthesis to genome replication (*Romero-López et al., 2014*; *Shetty, Stefanovic & Mihailescu, 2013*; *Tuplin et al., 2012*). The identification of the structures that form the core of these 'switches', and the dissection of the underlying molecular mechanism by which they work provides important insights into the replication of HCV and, by extrapolation, related viruses.

The HCV 5′UTR contains RNA signals essential for translation and replication and is an extensively structured region containing four stem-loop domains (SLI-IV; RNA structure naming conventions are detailed in Materials and Methods). Domains SLII-IV of the 5′UTR, along with the first 12–30 nucleotides of the core protein coding sequence, form the internal ribosome entry site (IRES) involved in cap-independent initiation of viral translation (*Friebe et al., 2001*; *Reynolds et al., 1995*). Two-additional structures within the core-coding region, SLV and SLVI, have also been implicated as important RNA elements in viral replication (*McMullan et al., 2007*). SLVI is a short stem-loop consisting of 54 paired bases, two sub-terminal bulge loops and a terminal loop of six nucleotides (*Tuplin, Evans & Simmonds, 2004*). In addition to forming functional RNA structures, the 5′UTR provides a platform for recruitment of the liver-specific microRNA 122 (miR122). In contrast to the typically repressive roles of cellular miRNAs, binding of miR122 to two seed sites (S1 and S2) near the 5′ end of the 5′UTR is critical for replication of HCV (*Jangra, Yi & Lemon, 2010*; *Jopling et al., 2005*; *Jopling, Schütz & Sarnow, 2008*), as well as stabilising the genome and providing protection against degradation (*Li et al., 2013*; *Sedano & Sarnow, 2014*; *Shimakami et al., 2012*).

'

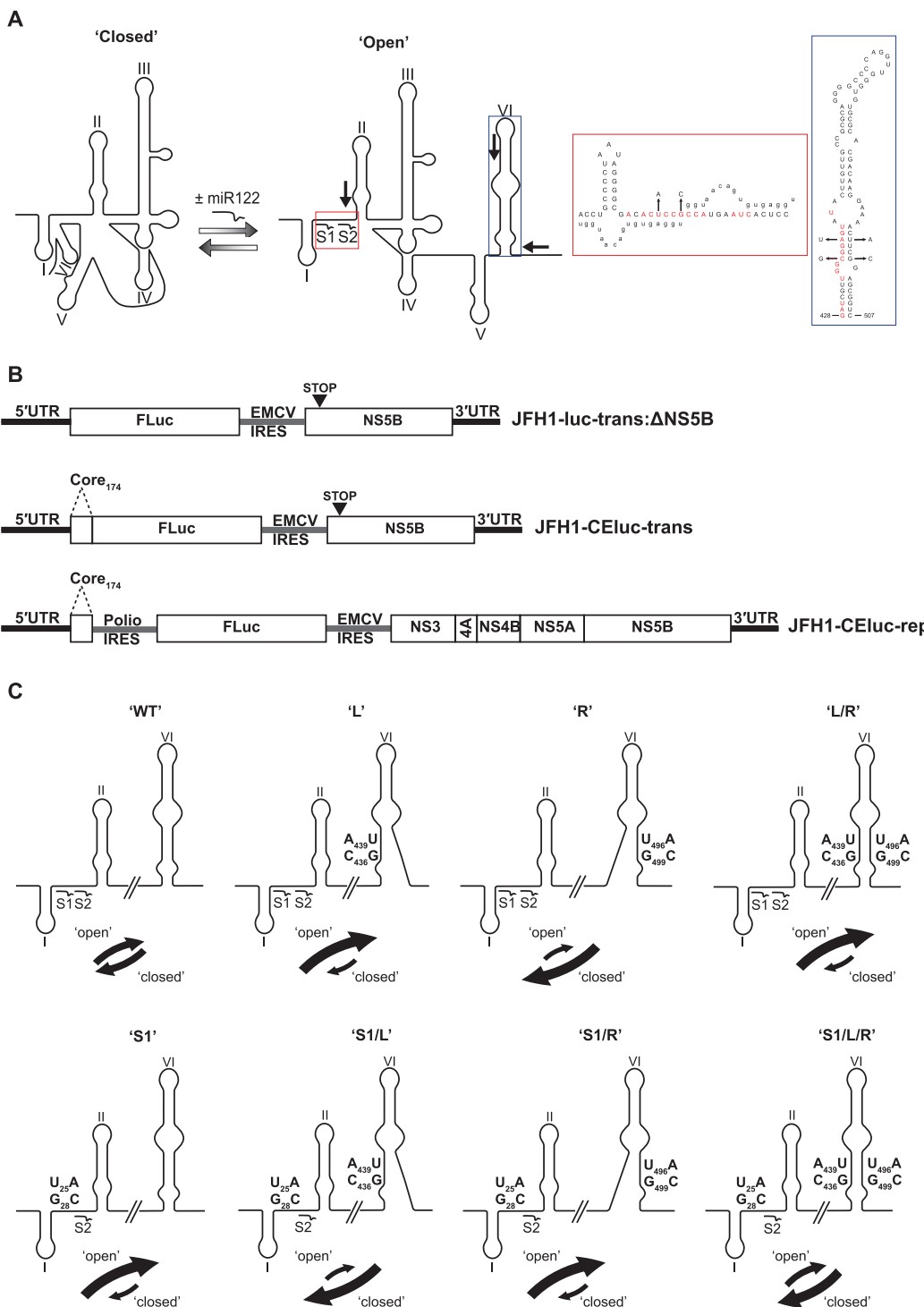

**Figure 1 Schematics of RNA structures and templates used.** (A) The predicted 'open' and 'closed' conformations of the HCV 5′UTR from SL I-VI with the addition, or loss, of miR122 as shown. S1 and S2 highlight the known binding sites of miR122. Black arrows indicate position and directionality of SHAPE primers. The red box shows an expanded view of nucleotides 1–42 of JFH-1 with miR122 (lowercase) bound to sites S1 and S2, with the blue box showing expanded view of nucleotides 428–507 of JFH-1 encompassing SLVI. Nucleotides in red are those predicted to be directly involved in formation of the LRA and mutations are indicated by faint black arrows indicating the substitutions made. Figure recreated and

adapted from (*Díaz-Toledano et al., 2009*). (B) The JFH-1 bicistronic translation reporter (JFH1-luc-trans:ΔNS5B), the core-extended translation reporter (JFH1-CEluc-trans), and replicon (JFH1-CEluc-rep). (C) The miR122 binding sites and SLVI with 5′ stem (L), 3′ stem (R) and S1 mutations displayed. The predicted blocking of miR122 binding, or SLVI formation, of each pair of mutations is shown. Black arrows represent the relative likely conformation, 'open' or 'closed', predicted to be favoured by each template.

Intriguingly, a sequence spanning the miR122 seed sites has also been demonstrated to anneal to complementary sequences that form the basal stem of SLVI. Mutation analysis of sequences in these regions had previously suggested a role for such an interaction in controlling translation (*Honda et al., 1999*; *Kim et al., 2003*). Subsequently, several methodologies, including RNase cleavage assays and atomic force microscopy, have been used to probe this region and map conformational changes in RNA structure involving these sequences (*Beguiristain, Robertson & Gómez, 2005*; *Díaz-Toledano et al., 2009*; *García-Sacristán et al., 2015*). Consequently, as identified by *Díaz-Toledano et al. (2009)*, a conformational change is induced via the three-way interplay of miR122, the 5′UTR and SLVI and this may function as a molecular switch, regulating translation and replication. In the simplest scenario, two mutually-exclusive, higher-order RNA structures are predicted as possible; 'open', in which miR122 is bound to the 5′UTR, and the 'closed' structure, in which there is a long-range association (LRA), between the miR122-binding site in the 5′UTR and the base of SLVI in the core-coding region (Fig. 1A).

In this study, we have utilised a panel of defined 5′UTR and SLVI mutations to investigate a translation-modulating role for the switch from the 'closed' to the 'open' conformation of this potential molecular switch. Selective 2′-hydroxyl acylation analysed by primer extension (SHAPE; *Merino et al., 2005*) analysis confirms previous reports that the change from the 'closed' to the 'open' conformation is influenced by the availability of miR122 (*Díaz-Toledano et al., 2009*). We further show that the 'closed' conformation is only achieved in the absence of miR122, or when both miR122 binding is blocked and base pairing within the basal stem of SLVI is prevented. Importantly, we were unable to correlate the 'open' or 'closed' conformations with specific translation phenotypes in either the presence or absence of miR122, and replication of the virus genome was apparently unaffected by these gross structural changes. Therefore, whilst the presence of SLVI is undoubtedly important for regulation of translation, the LRA between the miR122 binding sites in the 5′UTR and the base of SLVI, necessary for the formation of the 'closed' conformation, may not have an important regulatory role in either HCV replication or translation.

# MATERIALS AND METHODS

## Cell culture and transfection

Huh 7.5 human hepatocellular carcinoma cells were maintained at 37 °C, 5% $CO_2$, in Dulbecco's modified minimal essential medium (DMEM) (Sigma-Aldrich, St. Louis, MO, USA) supplemented with 10% (v/v) heat-inactivated foetal bovine serum (FBS), 1% non-essential amino acids and 2 mM L-glutamine (DMEM-FBS; GIBCO, Life Technologies, Carlsbad, CA, USA). HeLa cells were maintained at 37 °C, 5% $CO_2$,

in DMEM supplemented with 10% (v/v) FBS. Cells were seeded in 24-well plates at $1 \times 10^5$ cells/well for translation assays, or $0.8 \times 10^5$ cells/well for replicon assays, in DMEM-FBS 18 h prior to transfection.

Transfections were carried out with 500 ng RNA and two µl of Lipofectamine 2000 (Life Technologies, Carlsbad, CA, USA) as per manufacturers' instructions. In addition, for translation assays five ng of a capped and polyadenylated Renilla luciferase RNA was added as a transfection control. Transfection media was replaced at 4 h with fresh DMEM-FBS. Cell lysates were harvested for analysis at 4 h (translation assay) or 4, 21, 28 and 45 h (replicon assay). Briefly, cells were washed twice with PBS and lysed with 0.1 ml/well Glo Lysis buffer (Promega, Madison, WI, USA) for 15 min with shaking. Luciferase readings were determined with Dual-Luciferase (translation assays), or Luciferase (replicon assays), Assay System Kits (Promega, Madison, WI, USA) as per manufacturers' instruction.

## HCV cDNA plasmids, JFH-1 replicon construction and mutagenesis

A variety of numbering or naming schemes have been used to specify RNA stem-loop structures in the HCV genome. Although the most logically extendable scheme involves numbering structures according to the location of the 5′ nucleotide within the relevant genome region in an H77 reference sequence (*Kuiken et al., 2006*), for consistency with previous work on the HCV IRES and long-range structures between coding and non-coding regions, we use the SLI-VI scheme here. For comparison, SLVI is SL87 according to the *Kuiken et al. (2006)* nomenclature, due to the 5′ most nucleotide being located at nt 87 of the core-coding region.

A JFH-1 based translation-only reporter construct containing a core-extended (CE) sequence, to include SLV and VI—designated pJFH1-CEtrans—was generated via modification of pJFH1-luc-trans:ΔNS5B described previously (*Tuplin et al., 2015*). Overlap PCR was used to generate a DNA consisting of the 5′ end of JFH-1, including the first 174 nt of JFH-1 core, fused to the 5′ end of Firefly luciferase. An *Age*I/*Xba*I fragment was ligated into similarly digested pJFH1-luc-trans:ΔNS5B to generate pJFH1-CEtrans.

Mutations at miR122 seed site 1 (S1) or/and SLVI were introduced into pJFH1-CEtrans using the QuikChange II site-directed mutagenesis kit as per manufacturers' instructions (Agilent, Santa Clara, CA, USA). To disrupt miR122 binding, point mutations $U_{25}A$ and $G_{28}C$ were introduced into the miR122 S1 site to generate pJFH1-CEtrans-S1 (F:5′-GGCGACACACCCCCATGAATCACTC-3′ and R:5′-GAGTGATTCATGGGGGTG TGTCGCC-3′). To disrupt the SLVI structure point mutations $C_{436}G$ and $A_{439}U$ (F:5′-CCAG ATCGTTGGGGGTTATACTTGTTGC-3′ and R:5′-GCAACAAGTATACACCCCCAACG ATCTGG-3′), and/or $U_{496}A$ and $G_{499}C$ (F:5′-CGACAAGGAAAACATCCGAGCGG TCCCAGC-3′ and R:5′-GCTGGGACCGCTCGGATGTTTTCCTTGTCG-3′), were introduced into the 5′ and/or 3′ stem of SLVI to generate pJFH1-CEtrans-L, pJFH1-CEtrans-R and pJFH1-CEtrans-L/R, respectively. Mutants pJFH1-CEtrans-S1/L, pJFH1-CEtrans-S1/R and pJFH1-CEtrans-S1/L/R, containing both the miR122 S1 and SLVI mutations were generated by site-directed mutagenesis of pJFH1-CEtrans-L,

pJFH1-CEtrans-R and pJFH1-CEtrans-L/R with the pJFH1-CEtrans-S1 primers described above. All mutations were confirmed by sequence analysis.

A JFH-1 replicon, containing the extended core sequence as above, and designated pJFH1-CErep, was designed based on Con1b-luc-rep (*Tuplin et al., 2015*). Overlap PCR was used to replace the Con1b 5′UTR with that of JFH-1 within Con1b-rep-luc to generate pJFH1-5′UTR-Con1b-rep. A second overlap PCR generated a DNA containing the 3′ end of Firefly luciferase, the EMCV IRES and an ATG codon for NS3, and introduced into the previously described pJ6/JFH-1 (*Lindenbach et al., 2005*) to generate pJFH1-EMCV. An *Sbf*I/*Not*I fragment from pJFH1-5′UTR-Con1b-rep containing the JFH-1 5′UTR, poliovirus IRES and Firefly luciferase, was inserted into similarly digested pJFH1-EMCV to generate pJFH1-rep. An *Sbf*I/*Pme*I fragment of pJFH-1-CEtrans was inserted into similarly digested pJFH1-rep to generate the core-extended JFH-1 replicon, pJFH1-CErep. Mutations at miR122 S1 and/or SLVI were introduced through *Sbf*I/*Pme*I digestion of the appropriate pJFH1-CEtrans plasmid, and ligation into pJFH1-CErep. This resulted in replicon constructs pJFH1-CErep-S1, pJFH1-CErep-L, pJFH1-CErep-R, pJFH1-CErep-L/R, pJFH1-CErep-S1/L, pJFH1-CErep-S1/R and pJFH1-CErep-S1/L/R.

## miR122 duplexing and electrophoretic mobility shift assay

Native (miR122), complementary (miR122-Comp) and S1 mutant (S1-miR122) miR122 RNAs (5′-UGGAGUGUGACAAUGGUGUUUGU-3′, 5′-AAACGCCAUUAUCACA CUAAAUA-3′ and 5′-GGGUGUGUGACAAUGGUGUUUGU-3′) were synthesized by Integrated DNA Technologies along with an RNA oligonucleotide corresponding to nucleotides 1–50 of the JFH-1 strain of HCV (JFH1$^{1-50}$).

For addition of miR122 to replicon assays, miR122 RNA (10 mM) was duplexed with miR122-Comp (10 mM) in a final concentration of 100 mM HEPES, five mM $MgCl_2$, heated at 65 °C for 5 min and cooled slowly to room temperature.

For electrophoretic mobility shift assays (EMSA) JFH1$^{1-50}$ RNA (10 pmol) was heated for 5 min at 65 °C followed by cooling to 35 °C for 1 min in a 5 µl volume containing 100 mM HEPES (pH 7.6), 100 mM KCl and five mM $MgCl_2$. miR122 RNAs were added at molar ratios of 0, 0.5, 1, 1.5, 2, 3 and 5 and incubated for a further 30 min at 37 °C. An equal volume of loading dye (30% glycerol, 0.5× TBE and five mM $MgCl_2$) was added and RNA complexes separated by non-denaturing gel electrophoresis (15% 29:1 acrylamide:bisacrylamide, 0.5× TBE and 5 mM $MgCl_2$) at 150 V, 4 °C, 3 h using a BioRad MiniProtean III gel system. Gels were stained with SYBR Gold (Life Technologies, Carlsbad, CA, USA) and RNA visualized using a Typhoon FLA 9500 (GE Healthcare, Chicago, IL, USA).

## In vitro RNA transcription

RNA transcripts were synthesized using a HiScribe™ T7 High Yield RNA Synthesis Kit (NEB), as per manufacturers' instructions, with one µg of DNA template linearized with *Bsp*HI (for translation templates) or terminated with a 3′ *cis*-acting ribozyme from an *Mlu*I linearized template (for replicon templates). Newly transcribed RNA was

column-purified using a GeneJET RNA Purification Kit (Thermo Fisher Scientific, Waltham, MA, USA).

## RNA modification for SHAPE

A total of 10 pmol of translation construct-derived RNA transcripts were prepared in 10 μl of 0.5× Tris-EDTA (pH 8.0) (TE), denatured at 95 °C for 3 min and incubated on ice for 3 min prior to addition of six μl of either a five mM or 10 mM $MgCl_2$ folding buffer [333 mM HEPES (pH8.0), 333 mM NaCl and 16.5 mM or 33 mM $MgCl_2$]. Samples were allowed to refold at 37 °C for 20 min before being divided in half and incubated with either one μl of 100 mM $N$-methylisatoic anhydride (NMIA) dissolved in DMSO, or one μl of DMSO, at 37 °C for 45 min. For reactions in the presence of miR122 a three molar excess of miR122 was added prior to addition of folding buffer. Modified RNAs were column-purified using a GeneJET RNA Purification Kit (Thermo Fisher Scientific, Waltham, MA, USA) to remove miR122 prior to reverse transcription (RT).

## 5′-[$^{32}$P]-primer labelling

A total of 60 μM of primer was incubated with 10 units of T4 polynucleotide kinase (NEB), two μl of supplied 10× buffer and 10 μl $\gamma$-[$^{32}$P]-ATP ($3.7 \times 10^6$ Bq; PerkinElmer, Waltham, MA, USA) at 37 °C for 30 min followed by heat inactivation at 65 °C for 20 min. Radiolabelled primers were purified by separation on Sephadex G-25 Quick Spin Oligo Columns (Roche, Welwyn Garden City, UK).

## Primer extension for SHAPE

$N$-methylisatoic anhydride- or DMSO-treated RNA in 0.5× TE (10 μl) was mixed with three μl of radiolabelled primer, denatured at 95 °C for 5 min, annealed at 35 °C for 5 min and chilled on ice for 2 min. RT mix (six μl) was added (5× First Strand Buffer, five mM DTT, 0.5 mM dNTPs; Life Technologies, Carlsbad, CA, USA) and samples incubated at 55 °C for 1 min prior to addition of one μl of SuperScript®III (Life Technologies, Carlsbad, CA, USA) and further incubation at 55 °C for 30 min. The RNA template was degraded by addition of one μl of 4M NaOH and incubation at 95 °C for 5 min before addition of 29 μl of acid stop mix (140 mM un-buffered Tris-HCl, 73% formamide, 0.43× TBE, 43 mM EDTA [pH 8.0], bromophenol blue and xylene cyanol dyes) and further incubation at 95 °C for 5 min. Dideoxynucleotide (ddNTP) sequencing markers were generated by the extension of unmodified RNA with addition of two μl of 20 mM ddNTP (TriLink BioTechnologies, San Diego, CA, USA) prior to addition of RT mix. The cDNA extension products were separated by denaturing electrophoresis (7% (19:1) acrylamide:bisacrylamide, 1× TBE, 7M urea) at 70 W for 3–5 h depending on product sizes to be analysed. Gels were visualised with a phosphorimager (Typhoon FLA 9500) and densitometry analysis carried out with ImageQuant TL 8.1 software (GE Healthcare Life Sciences). Normalised reactivities indicating exposure of nucleotides in predicted RNA structures were calculated as described previously (*Tuplin et al., 2012*).

## RESULTS

### Mutagenesis of a miR122 binding site and the sequences implicated in the LRA

The previously probed 'open' and 'closed' structural conformations are determined by complementarity between the 5′ nucleotides forming the miR122 binding site and the base of SLVI in the core protein-coding region, together with the presence of exogenous miR122 (*Beguiristain, Robertson & Gómez, 2005*; *Díaz-Toledano et al., 2009*). The latter, by binding to the 5′UTR sequences, inhibits the LRA and 'opens' the structure (Fig. 1A). This transition from a 'closed' to an 'open' structure can be predicted bioinformatically using mfold (*Zuker, 2003*) and bifold RNA secondary structure prediction software (*Reuter & Mathews, 2010*), to demonstrate that if the S1 site is occupied by miR122 the 'open' conformation with bound miR122 is energetically more favourable (Fig. S1).

To investigate the existence and potential functions of the alternative conformations we first modified our existing JFH-1 translation reporter vector, JFH1-luc-trans:$\Delta$NS5B (Fig. 1B), to generate a core-extended version, JFH1-CEtrans, which encompasses the first 174 nucleotides of the core-coding region, thus incorporating SLVI. A JFH-1 based sub-genomic replicon, designated JFH1-CErep, was additionally constructed to include the same core-extended sequence (Fig. 1B). For both JFH1-CEtrans and JFH1-CErep we subsequently undertook a systematic mutagenesis of either, or both, of the complementary sequences required for formation of the LRA.

First, using mfold structure prediction (*Zuker, 2003*), we identified two sites within SLVI at which synonymous substitutions could be introduced that should disrupt formation of the structure (Fig. 1A). Substitutions $C_{436}G$ and $A_{439}U$ in the 5′ stem of SLVI (designated 'L' mutants) and $U_{496}A$ and $G_{499}C$ in the 3′ stem of SLVI (designated 'R' mutants) independently prevented base pairing of the basal stem of SLVI. Both $C_{436}$ and $A_{439}$ are implicated in the formation of the 'closed' structure and consequently, 'L' mutants were predicted to additionally prevent the LRA due to disruption of the complementarity with the miR122 seed site 1 (S1). Conversely, 'R' mutants would free the 5′ sequences forming the basal stem of SLVI to contribute solely to formation of the 'closed' structure. However, since formation of the 'closed' structure would also be dependent on the S1 site being unoccupied by miR122, we also introduced substitutions into the latter (at positions $U_{25}A$ and $G_{28}C$) that were predicted to prevent miR122 binding and at the same time would restore complementarity with the 'L' mutations in SLVI (Fig. 1C; Table 1).

To verify that binding of miR122 to S1 was abrogated in the S1 mutants we conducted EMSAs using synthetic miR122 and an RNA oligonucleotide corresponding to the first 50 nucleotides of JFH-1 ($JFH1^{1-50}$). With the addition of miR122 to unmodified $JFH1^{1-50}$ we observed the expected two complexes with reduced mobility, representative of binding of miR122 to both S1 and S2 seed sites. Saturation of both seed sites was achieved upon addition of a 3:1 molar ratio of miR122:$JFH1^{1-50}$ (Fig. 2A), while addition of an antisense miR122 RNA (miR122-Comp) showed no change in mobility (Fig. 2B). In contrast to unmodified $JFH1^{1-50}$, S1-mutated $JFH1^{1-50}$ only formed the faster

**Table 1 Substitutions and predicted conformations.**

| Mutant | Mutations | Predicted conformation[a] |
|---|---|---|
| Parental JFH1-CEtrans or JFH1-CErep | n/a | Open |
| JFH1-CEtrans-L/JFH1-CErep-L | $C_{436}G$, $A_{439}U$ | Open |
| JFH1-CEtrans-R/JFH1-CErep-R | $U_{496}A$, $G_{499}C$ | Closed |
| JFH1-CEtrans-L/R/JFH1-CErep-L/R | $C_{436}G$, $A_{439}U$, $U_{496}A$, $G_{499}C$ | Open |
| JFH1-CEtrans-S1/JFH1-CErep-S1 | $U_{25}A$, $G_{28}C$ | Open |
| JFH1-CEtrans-S1/L/JFH1-CErep-S1/L | $U_{25}A$, $G_{28}C$, $C_{436}G$, $A_{439}U$ | Closed |
| JFH1-CEtrans-S1/R/JFH1-CErep-S1/R | $U_{25}A$, $G_{28}C$, $U_{496}A$, $G_{499}C$ | Open |
| JFH1-CEtrans-S1/L/R/JFH1-CErep-S1/L/R | $U_{25}A$, $G_{28}C$, $C_{436}G$, $A_{439}U$, $U_{496}A$, $G_{499}C$ | Closed |

Note:
[a] In the presence of miR122.

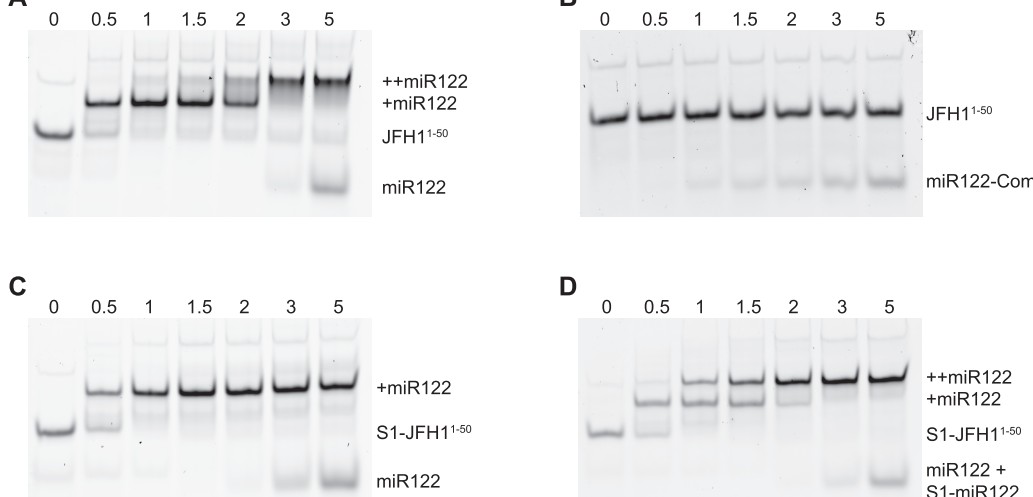

**Figure 2 RNA-RNA electrophoretic mobility gel shift assays of miR122 binding to JFH-1 5′UTR.**
A synthetic RNA of nts 1–50 of JFH-1 ($JFH1^{1-50}$) was complexed with increasing molar ratios of (A) wild type or (B) antisense synthetic miR122 and separated by non-denaturing PAGE. $JFH1^{1-50}$ mutated at the S1 binding site was similarly complexed with (C) wild type or (D) wild type plus an S1-mutated miR122 and separated by non-denaturing PAGE. miR122 binding was denoted by 1 (+ miR122) or 2 (++ miR122) reductions in RNA mobility compared to $JFH1^{1-50}$ control with no miR122 present (lane 1, A–D).

migrating single complex, even at a 5:1 molar ratio, indicating that miR122 remained bound to S2 alone (Fig. 2C). Restoration of both mobility-shifted complexes was achieved upon addition of a 50–50 mix of unmodified and S1-modified miR122 (S1-miR122), the latter containing mutations complementary to those introduced in S1 mutated $JFH1^{1-50}$ (Fig. 2D). These studies confirmed that substitutions introduced to the S1 site were sufficient to disrupt miR122 binding to the S1 seed site, but that binding to the S2 seed site was unaffected, in agreement with similar mutation analysis of miR122 binding (*Mortimer & Doudna, 2013*).

To investigate the influence on the conformation of the 5′ end of the HCV RNA the L, R and S1 mutations predicted to influence the 'open' or 'closed' conformation were

introduced individually, or in combination, into the core-extended translation and replicon reporters, JFH1-CEtrans and JFH1-CErep, respectively, and individual templates validated by sequence analysis (Table 1).

## The LRA is detected in the absence, but not presence, of miR122

We have previously used SHAPE mapping to demonstrate a long-range interaction between the 3′UTR of the HCV genome and distal sequences located within the polyprotein-coding region (*Tuplin et al., 2012*). These interactions occurred only in cis and were acutely sensitive to point mutations within the complementary regions. We were therefore confident SHAPE analysis could provide useful insights into the study of the LRA. Three regions of the HCV RNA were analysed to provide data on RNA structure: (1) the 5′ base stem of SLVI, (2) the 3′ base stem of SLVI and, (3) nts 1–80 of the 5′UTR. Unfortunately, the presence of a highly stable stem-loop (SLI; Fig. 1A) immediately 5′ to the S1 miR122 binding site acted as a strong terminator during cDNA synthesis. Consequently, as others have previously found (*Pang et al., 2012*), the 5′ end of the S1 miR122 binding site (nts 1–20) proved difficult to accurately map due to excessive background signal. Scrutiny of the predicted pattern of base pairing between the miR122 binding site and miR122 also shows that it is highly similar to that between the miR122 binding site and the 5′ base stem of SLVI. Together, these issues meant that the S1 region was not informative for defining the 'open' or 'closed' conformation. Determination of the 'closed' structure was therefore based primarily on the structure of SLVI. In particular, nucleotides 434–435 (GG), which are predicted to be paired when involved in the LRA but unpaired in formation of the basal stem of SLVI, and the overall paired/unpaired nature of the 3′ side of the basal stem of SLVI (nucleotides 494–507), which would be predominantly unpaired upon formation of the 'closed' structure (Fig. 1A and Fig. S1). Preliminary experiments showed that SHAPE mapping of the 5′ regions of a variety of unmodified templates, for example, JFH1-CEtrans, JFH1-CErep or a full-length in vitro transcribed RNA, resulted in the same NMIA reactivity and resulting structural predictions (data not shown). We conclude from this that the LRA interactions are essentially local in nature and are unaffected by distal sequences in the virus genome. All subsequent SHAPE mapping was conducted using JFH1-CEtrans as template.

We first compared the structural conformations of parental JFH1-CEtrans in the absence of miR122 during the RNA folding reaction, or with a 3:1 molar excess of miR122 to saturate binding to S1 and S2, as determined from EMSAs (Fig. 2A). In the presence of miR122 the basal stem of SLVI was predominantly NMIA-unreactive, indicating that the pairing through this region was in agreement with the structure predicted bioinformatically (Fig. 3A). Indeed, the reactivity of nucleotides 427–447 and 487–507 corresponded very well with both the predicted and RNAse-probed structure of this region of SLVI (*Tuplin, Evans & Simmonds, 2004*). As additional validation we determined the NMIA-reactivity of SLVI sequences in JFH1-CEtrans in the presence of a locked-nucleic-acid (LNA) probe, J22. LNA J22 binds with high affinity to nts 21–37 of JFH-1 across the miR122 binding sites, allowing for the determination of the SLVI

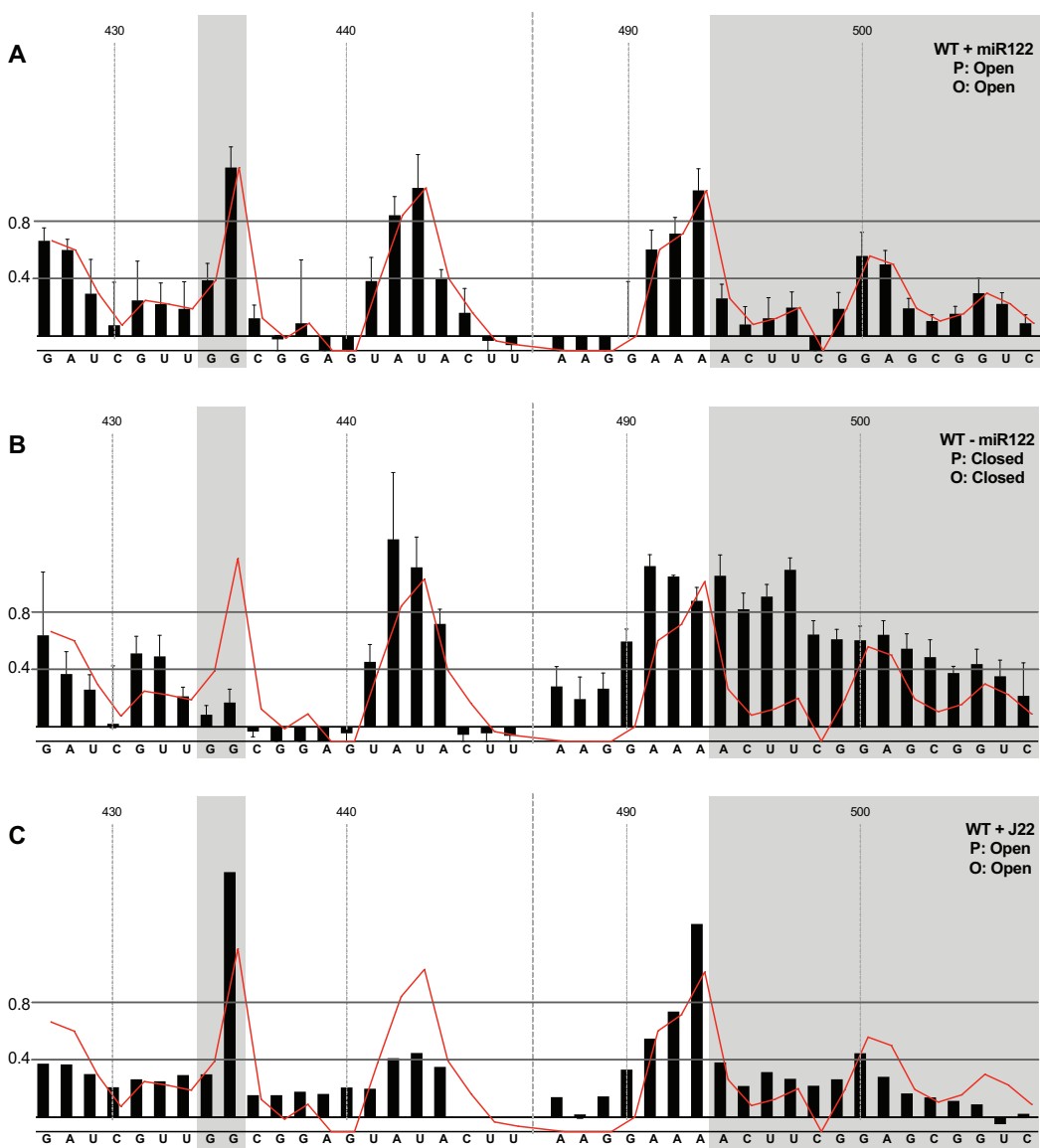

**Figure 3 SHAPE analysis of parental template plus/minus miR122.** SHAPE reactivities are shown for (A) JFH1-CEtrans plus miR122, (B) JFH1-CEtrans minus miR122, and (C) JFH1-CEtrans plus LNA J22, with predicted (P) and observed (O) conformations given top right below template name. Black bars show normalised SHAPE reactivities of nucleotides 427–447 and 487–507, encompassing the 5′ and 3′ basal stems of SLVI, respectively. Nucleotides with a reactivity of <0.4 are considered unreactive and therefore base-paired. Shaded regions highlight nucleotides of importance in determining 'open' or 'closed' conformations: specifically the 5′ $G_{434}G_{435}$ motif and 3′ nucleotides 494–507. The superimposed red line indicates the exposure of JFH1-CEtrans plus miR122, and is included on all plots for comparison of reactivities between a demonstrated 'open' conformation and the observed reactivity of additional templates. A maximum negative reactivity was set at −0.1. Unless otherwise stated, error bars represent the SD of a minimum of two independent gel analyses for two replicate RNA–NMIA folding reactions. Figure 3C was derived from only one replicate folding reaction.

structure independently of the reversible action of miR122 (Fig. 3C). The resulting SHAPE analysis recapitulated the results observed in the presence of miR122, with little or no reactivity of sequences known to form the basal stem of SLVI. These results support the

previously probed structure of SLVI indicating that, in the presence of miR122, the 'open' conformation predominates.

microRNA 122 is present at high levels in hepatic cells in which HCV replicates (*Lagos-Quintana et al., 2002*). Nevertheless, since there might be compartmentalisation—in replication complexes, for example—where miR122 is limited or absent, we went on to investigate the potential formation of the 'closed' structure by SHAPE in the absence of miR122 (Fig. 3B). Under these conditions we observed gross changes to the structure of the basal region of SLVI. The $G_{434}G_{435}$ motif—predicted to be a key interaction with the S1 site—are highly unreactive, indicating that they are base paired. At the same time, the reactivity of the 3′ sequences of the basal stem of SLVI increases. There are significant increases in exposure of nt 490–503 indicative of a more extensive opening out of the SLVI structure. We interpret this as the formation of the 'closed' structure in the absence of miR122, despite the inability to measure the reactivity of nucleotides within the S1 site. In contrast to the results of *García-Sacristán et al. (2015)*, but perhaps indicative of differences between in vitro techniques, we were unable to demonstrate a magnesium-dependent preference for the formation of the 'closed' structure while in the presence of miR122. We investigated the structure of the basal stem of SLVI in the parental templates at an increased concentration of 10 mM $MgCl_2$ and determined that the 'open' conformation predominated, irrespective of the magnesium concentration (Fig. S2).

Together, these results are in agreement with a previous conclusion by *Díaz-Toledano et al. (2009)* obtained via RNase III cleavage assays, and are highly indicative of an inhibitory role for miR122 in formation of the 'closed' structure.

## In the presence of miR122, the LRA is favoured only when both miR122 binding and the SLVI structure are disrupted

Having investigated the pairing of the basal stem of SLVI and the occurrence of the LRA in unmodified templates, we went on to study the influence of mutations introduced to prevent these interactions, or that we had previously shown prevent miR122 binding. All subsequent analyses were carried out in the presence of miR122.

We first analysed those templates predicted to preferentially form the 'open' conformation (Fig. 4). Modification of the S1 site in template JFH1-CEtrans-S1 (Fig. 4A), shown to abrogate miR122 binding (Fig. 2C), resulted in a NMIA-reactivity pattern almost indistinguishable from the unmodified parental template (compare Fig. 3A with Fig. 4A). Since the $U_{25}A$ and $G_{28}C$ changes in the S1 mutant also prevents interaction with SLVI nts $A_{439}$ and $C_{436}$, respectively, this provides further support that this pattern of NMIA reactivity represents the 'open' conformation. Three additional modified templates, JFH1-CEtrans-L, JFH1-CEtrans-L/R and JFH1-CEtrans-S1/R, were also predicted to block the LRA by reducing the complementarity between nucleotides in the S1 site and the basal stem of SLVI (Fig. 1C). NMIA-reactivity of these three templates verified the predicted inhibition of the LRA, as evidenced by the high reactivity of the $G_{434}G_{435}$ motif (Figs. 4B, 4C and 4D). Interestingly, in comparison to the parental JFH1-CEtrans, all three templates displayed increases in reactivity in other regions,

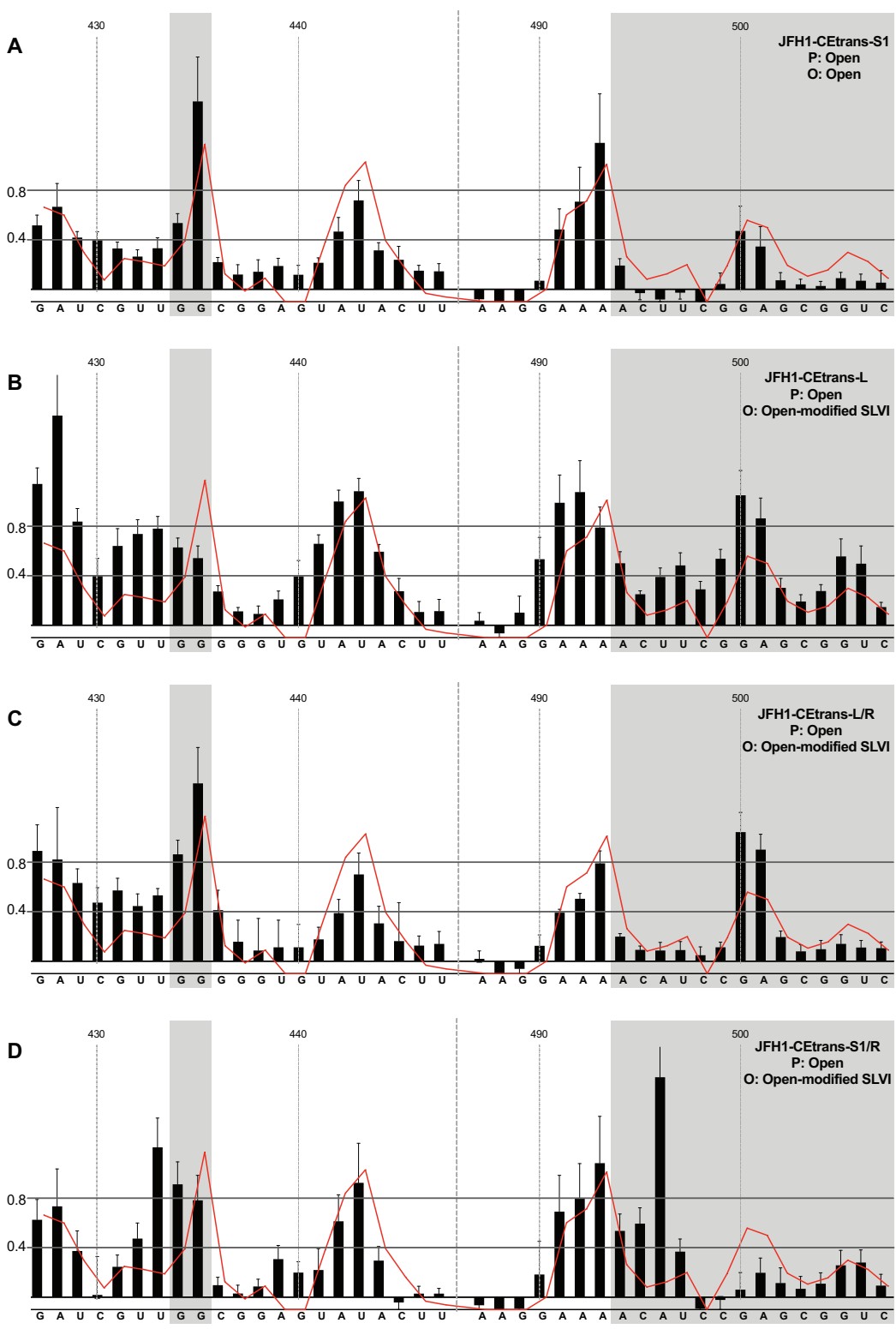

**Figure 4 SHAPE analysis of S1 and SLVI mutants with predicted 'open' conformation.** SHAPE reactivities are shown for (A) JFH1-CEtrans-S1, (B) JFH1-CEtrans-L, (C) JFH1-CEtrans-L/R and (D) JFH1-CEtrans-S1/R, with predicted (P) and observed (O) conformations given top right below template name. Data presentation as described in Fig. 3.

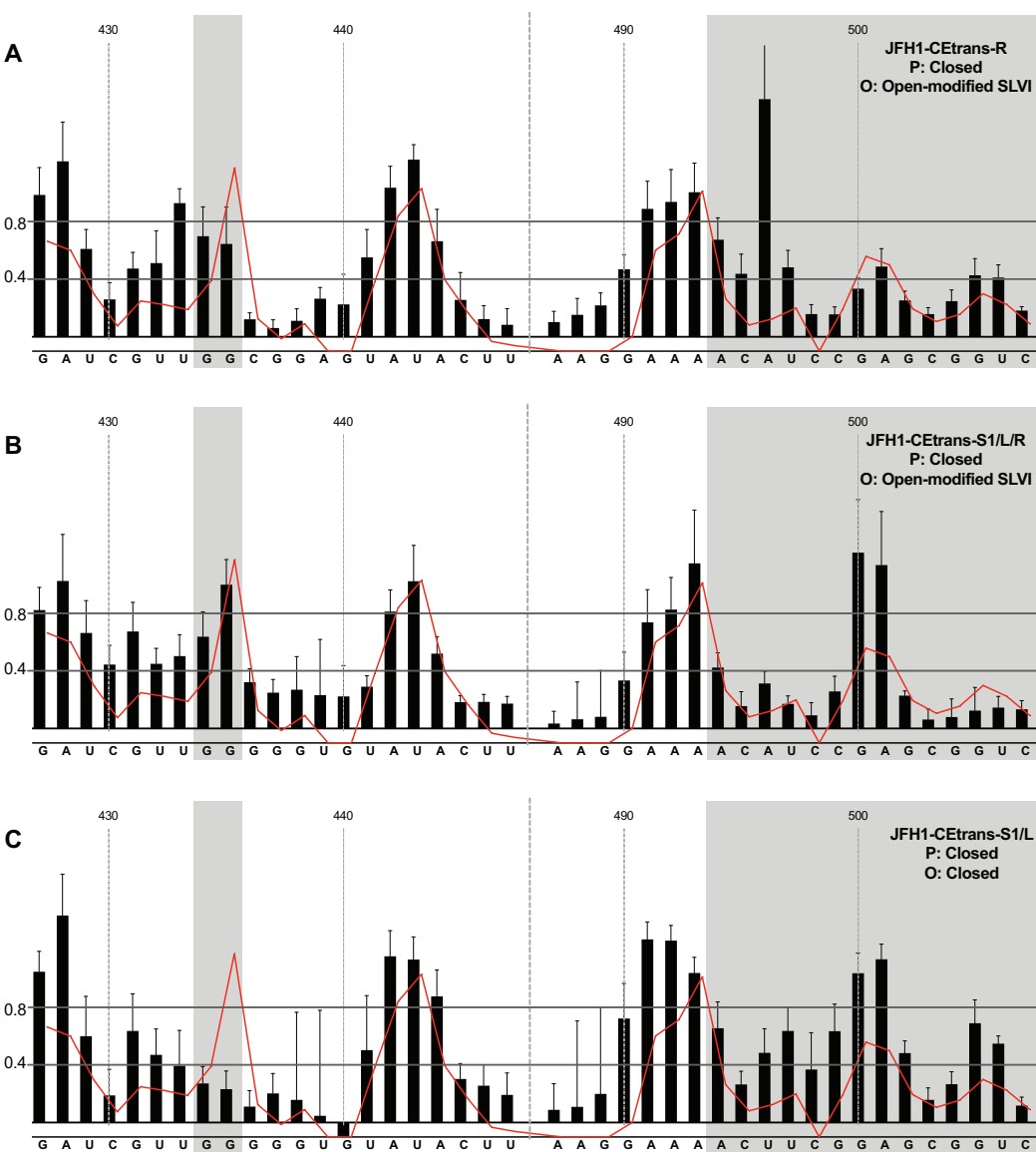

**Figure 5 SHAPE analysis of S1 and SLVI mutants with predicted 'closed' conformation.** SHAPE reactivities are shown for (A) JFH1-CEtrans-S1/L, (B) JFH1-CEtrans-R and (C) JFH1-CEtrans-S1/L/R, with predicted (P) and observed (O) conformations given top right below template name. Data presentation as described in Fig. 3.

such as nts 427–433 (the 5′ basal stem) or nts 491–495 (3′ basal stem), suggesting that, while preventing the LRA, the introduced mutations may also lead to generation of an altered SLVI structure (compare line graph to black bars in Figs. 4B, 4C and 4D). The mfold predictions for the structure of SLVI containing mutations $C_{436}G$ and $A_{439}U$, and $U_{496}A$ and $G_{499}C$, did not suggest formation of such an altered structure (data not shown) and it is not possible to deduce the precise structure from the NMIA-reactivity plots.

We next investigated the conformation of templates containing combinations of mutations that were predicted to favour the LRA and the 'closed'

conformation: JFH1-CEtrans-R, JFH1-CEtrans-S1/L and JFH1-CEtrans-S1/L/R (Fig. 5). Unexpectedly, both JFH1-CEtrans-R and JFH1-CEtrans-S1/L/R failed to demonstrate the LRA, again as evidenced by the reactivity of the $G_{434}G_{435}$ motif, as well as the overall lack of reactivity in the 3′ basal stem of SLVI that would be expected (Figs. 5A and 5B). As with JFH1-CEtrans-L and JFH1-CEtrans-L/R, we observed an overall increase in reactivity of the 5′ basal stem nucleotides suggestive of a similar disruption to the SLVI structure that was not predicted in mfold calculations (compare Figs. 5A and 5B with Figs. 4B and 4C). However, these result suggest that despite significant disruption to the known SLVI structure, the 'closed' structure is not the favoured conformation for the RNA template.

In contrast to all other modified templates JFH1-CEtrans-S1/L generated a NMIA-reactivity plot matching that of parental JFH1-CEtrans in the absence of miR122, and is highly indicative of the formation of the 'closed' structure (Fig. 5C). In comparison to a template in which the 'closed' structure is blocked, that is, parental JFH1-CEtrans in the presence of miR122, the mean NMIA-reactivities of the JFH1-CEtrans-S1/L $G_{434}G_{435}$ motif were reduced from 0.38 and 1.16 to 0.26 and 0.22, respectively, highlighting a substantial change in the base paired state, especially of $G_{435}$. Similarly, the average reactivity of nts 490–507 in the 3′ basal stem was increased from 0.28 to 0.64 demonstrating the overall increase in reactivity expected when the 5′ basal stem of SLVI is bound to the miR122 S1 site in the 'closed' conformation.

Taken together the SHAPE analyses show that, in the presence of miR122, the 'closed' conformation only exists when both miR122 binding at S1 is blocked, and nucleotide complementarity between S1 and the 5′ basal stem of SLVI is restored.

## Phenotypic characterisation of LRA-modified templates

Using SHAPE analyses we determined that, in the presence of miR122, the LRA resulting in the 'closed' conformation, is highly unlikely to form. However, if HCV translation and/or replication occur in locations or complexes in which miR122 is absent then the 'closed' structure is the energetically favourable conformation (Fig. 3B), and as such may influence virus translation and replication. We therefore used selected modified templates with demonstrated changes in conformation, to investigate the effects of the LRA on translation and replication.

Hepatitis C virus IRES-mediated translation is known to require the first 12–30 nt of the core protein-coding region (Reynolds et al., 1995). However, to study the effects on translation of the LRA required the additional SLV–SLVI sequences included in the core-extended translation reporter described above and utilised in SHAPE analysis (Fig. 1B). We initially compared translation from JFH1-luc-trans:ΔNS5B, containing the minimal core sequence, to the core-extended JFH1-CEtrans reporter. In agreement with previous observations (Kim et al., 2003), translation was significantly decreased (~2.5-fold) with the inclusion of the extended core sequence (Fig. 6A). This reduction in translation is proposed to be a result of formation of the LRA (Kim et al., 2003) and that a high proportion of the JFH1-CEtrans RNA templates exist in the 'closed' conformation (Fig. 1A), and hence are unavailable for use by the cellular translation

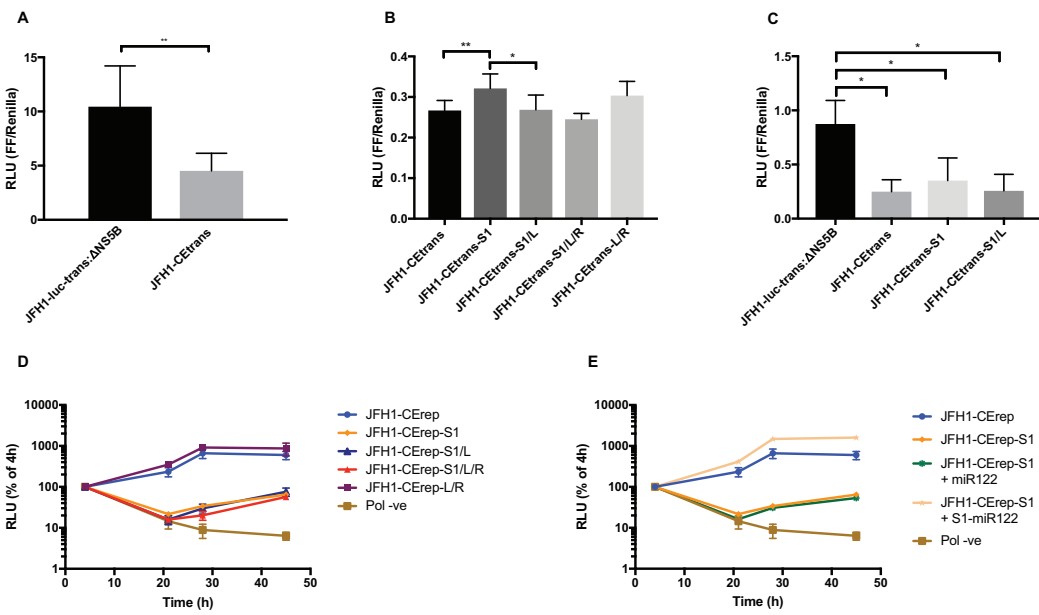

**Figure 6  Phenotypic characterisation of JFH-1 reporter bearing S1 and SLVI mutations.** Translation levels were determined by luciferase assay for (A) JFH1-luc-trans:ΔNS5B and JFH1-CEtrans, (B) JFH1-CEtrans, S1 and SLVI mutants and (C) JFH1-luc-trans:ΔNS5B and JFH1-CEtrans in HeLa cells. Cell lysates were harvested at 4 h and luciferase readings calculated as a ratio of Firefly luciferase to a co-transfected Renilla luciferase control RNA. Replication kinetics of (D) JFH1-CErep, S1 and SLVI mutants and (E) JFH1-CErep-S1 supplemented with S1-miR122, were determined by luciferase assays at 4, 21, 28 and 45 h post-transfection. Luciferase readings are expressed as a percentage of the 4 h reading to normalise against translation of input RNA. A polymerase active site mutant, GDD to GNN, was included as replication control (Pol −ve). For all assays error bars represent SD of three replicate transfections from triplicate experiments, with statistical significance calculated by unpaired t-test analysis using GraphPad Prism V7.

machinery. As we have demonstrated that the LRA occurs only in the absence of miR122 (Fig. 3), this result implies either the exclusion of miR122 from sites of translation or, an alternative role for the sequences encompassing SLV and SLVI domains in regulating translation. If the observed reduction is a result of the LRA, translation levels should be restored by mutations designed to disrupt the LRA, so forcing the 'open' conformation, and repressed again by compensatory substitutions that—although different from the parental template—restore the LRA and the 'closed' conformation.

We therefore compared translation from JFH1-CEtrans with selected modified templates that had shown a distinct conformation in SHAPE analysis. We selected JFH1-CEtrans-S1 as a representative of the 'open' conformation due to the SHAPE analysis most closely matching the SLVI structure observed for parental JFH1-CEtrans. For the 'closed' conformation we were limited to the study of JFH1-CEtrans-S1/L, as the only modified template in which the LRA was demonstrated. In addition, we investigated JFH1-CEtrans-L/R and JFH1-CEtrans-S1/L/R, for which the SHAPE reactivities suggested that the 5′ and 3′ stem mutations did not recapitulate wild type SLVI as expected due to increased reactivity in the 5′ basal stem, but neither were they representative of the 'closed' conformation (Figs. 4C and 5B). We transfected Huh 7.5 cells with 500 ng of RNA of each template, in parallel, and normalised translation levels to a

Renilla luciferase transfection control RNA (Fig. 6B). As predicted for the 'open' conformation and blocking of the LRA, JFH1-CEtrans-S1 showed a small, but significant, increase in translation level compared to both JFH1-CEtrans and JFH1-CEtrans-S1/L ($p < 0.05$). In contrast, JFH1-CEtrans-S1/L, shown to preferentially form the 'closed' structure, did not show the predicted reduction in translation and was unchanged from parental JFH1-CEtrans. Similarly, despite small increases and decreases in translation levels of JFH1-CEtrans-L/R and JFH1-CEtrans-S1/L/R, respectively, these were not significantly different when compared to the parental template. This suggests that, even though the wild type SLVI structure is not fully restored, with reduced base pairing remaining in the basal 5′ stem, this did not impact translation.

The limited changes in phenotype observed strongly suggested that the difference observed between JFH1-luc-trans:ΔNS5B and JFH1-CEtrans is due to an as yet unidentified role for SLV and VI, and is not miR122 or LRA dependent. In the presence of miR122 both these templates exhibit the 'open' conformation as determined by SHAPE analysis (Figs. 3A and 4A). Due to the mutations introduced into JFH1-CEtrans-S1 only JFH1-CEtrans is capable of forming the 'closed' conformation which, in the absence of miR122, is the preferred conformation (Fig. 3B). As the switch between the two conformations is miR122 dependent we reasoned that, if the 'closed' conformation is in fact responsible for the reduction in translation of JFH1-CEtrans, we would observe a similar difference when comparing JFH1-CEtrans ('closed') to JFH1-CEtrans-S1 ('open') in cells lacking miR122. We therefore investigated translation of JFH1-luc-trans:ΔNS5B, JFH1-CEtrans, JFH1-CEtrans-S1 and JFH1-CEtrans-S1/L in HeLa cells, which are naturally lacking in miR122 (Jopling et al., 2005) (Fig. 6C). Overall we observed translation levels that were ~10-fold lower in HeLa cells than in Huh 7.5 cells. However, the relative translation phenotypes remained the same, with a significant reduction in translation for JFH1-CEtrans, and all other templates that contained the core-extended sequence. Additionally, no significant changes were observed between JFH1-CEtrans and JFH1-CEtrans-S1, or JFH1-CEtrans-S1/L, which is also expected to form the 'closed' conformation in the absence of miR122. This supports our contention that formation of the LRA does not account for the change in translation phenotype between JFH1-luc-trans:ΔNS5B and JFH1-CEtrans, and conclude that—although the presence of SLVI clearly reduces translation (Figs. 6A and 6C)—this is unrelated to the LRA and any miR122 induced switch between the 'open' and 'closed' conformations.

With no observable link between translation levels and the conformations demonstrated by SHAPE analysis we went on to investigate whether there were differences in replication between our structurally modified templates. To achieve this we used a core-extended version of a JFH-1 replicon, JFH1-CErep. Huh 7.5 cells were transfected with 500 ng of template RNA, containing the same modifications as above, and replication recorded as luciferase activity over a 45 h time course. Our results show that for JFH1-CErep-L/R, where mutations were engineered solely within SLVI, replication was not significantly altered from the unmodified parental replicon, in keeping with the results for translation (Figs. 6B and 6D). In contrast, mutants JFH1-CErep-S1, JFH1-CErep-S1/L and JFH1-CErep-S1/L/R, in which miR122 binding to S1 is blocked, all showed a

>10-fold, highly significant reduction in the level of replication ($p < 0.01$) (Fig. 6D). This is entirely consistent with our current understanding that miR122 binding to the 5′UTR of HCV is important for HCV replication (*Jangra, Yi & Lemon, 2010*; *Jopling et al., 2005*). The addition of modified S1-miR122 to the RNA transfection mix, to complement the S1 mutations in the replicon, restored replication of JFH1-CErep-S1 to parental levels, demonstrating that the observed reductions in replication are due entirely to disruption of miR122 binding and not a consequence of possible changes in conformation promoted by the LRA (Fig. 6E).

These results demonstrate that, while miR122 binding has a profound effect on the replication of the HCV genome, neither replication nor translation phenotypes are significantly influenced by modifications to SLVI that preferentially form the 'open' or 'closed' conformations of the LRA.

## DISCUSSION

How are the competing events of single-stranded, positive-sense, RNA virus translation and replication separated? At least early in the replication cycle, before compartmentalization into membrane-bound replication vesicles, these must involve interaction of the translating ribosome or the viral polymerase with the same template. One strategy, typified by poliovirus, requires the accumulation of one or more viral translation products to initiate genome replication (*Barton & Flanegan, 1997*; *Jurgens & Flanegan, 2003*). In this case, ribonucleoprotein complexes form involving phylogenetically conserved RNA stem-loop structures. Since many positive-sense single-stranded RNA viruses have small genomes, the adoption of higher-order structures—that may vary in conformation depending upon the environment or availability of interacting proteins—effectively increases the level of control that can be exerted during the replication cycle. In particular, structures capable of forming long-range interactions are of interest as they may have the capability to cyclize the genome (*Alvarez et al., 2005*) or—by adopting alternate conformation—riboswitches (*Ooms et al., 2004*; *Shetty, Stefanovic & Mihailescu, 2013*; *Wang & White, 2007*).

The HCV genome is extensively structured (*Mauger et al., 2015*; *Simmonds, Tuplin & Evans, 2004*; *Tuplin et al., 2012*) and a number of long-range interactions have been predicted within it (*Fricke et al., 2015*), some having demonstrable involvement in important aspects of the HCV replication cycle (*Diviney et al., 2008*; *Romero-López et al., 2014*; *Romero-López & Berzal-Herranz, 2009*; *Shetty, Stefanovic & Mihailescu, 2013*; *Tuplin et al., 2015*; *You & Rice, 2008*). Of these, the LRA between nts 23–31 encompassing miR122 seed site 1 in the 5′UTR, and complementary nts 433–441 located in the 5′ basal stem of SLVI within the core protein-coding region (*Beguiristain, Robertson & Gómez, 2005*; *Díaz-Toledano et al., 2009*; *García-Sacristán et al., 2015*), may adopt 'open' and 'closed' conformations, and is predicted to modulate the switch between translation and replication of the virus genome (Fig. 1A) (*Kim et al., 2003*). The 'open' conformation (i.e. no LRA) is proposed to favour translation, whereas the 'closed' conformation restricts access of the ribosome to sequences within the coding region implicated in genome translation, thereby favouring replication.

To investigate the structure and function of the 'open' and 'closed' conformations in greater detail we have mapped the native structure using SHAPE in the presence and absence of miR122. We have additionally extensively mutagenised sequences implicated in formation of both the 'open' and 'closed' conformations, mapped their structure by SHAPE and investigated the resulting influence on translation and genome replication.

Although the strong stem-loop (SLI) in the 5′UTR confounded SHAPE interrogation of sequences forming the S1 site of miR122 binding, those contributing to the basal stem of SLVI were readily mapped. Having determined the influence on miR122 binding of mutations in the S1 site (Fig. 2) we inferred the LRA and formation of the 'closed' structure from exposure or otherwise of the basal stem of SLVI. The LRA was detectable only under very specific conditions, including an in vitro assay in which miR122 was omitted. Similarly, mutagenesis of the template within the S1 miR122 binding site (to prevent miR122 binding) and introduction of complementary mutations to the 5′ basal stem of SLVI allowed the LRA to be inferred. In contrast, in the presence of miR122, and/or unmodified sequences at the basal stem of SLVI, we were unable to detect the LRA and formation of the 'closed' structure. We propose that, under conditions in which miR122 is present in significant amounts, the phylogenetically conserved basal stem of SLVI is unlikely to separate to form a LRA.

However, incorporation of 5′ (L) and 3′ (R) mutations to SLVI did lead to structural changes within the stem-loop that were not predicted by mfold. With the exception of JFH1-CEtrans-S1/L, which clearly adopts the 'closed' conformation, all the tested substitutions to the basal stem of SLVI increased the NMIA-reactivity of the structure (Figs. 4 and 5), indicating a reduction in complementary pairing that was more extensive than the sites of modification. In addition, when not paired with the 5′ mutations, the 3′ mutants (JFH1-CEtrans-R and JFH1-CEtrans-S1/R) showed further modification of the SLVI structure with the loss of reactivity of nts 500–501 (Figs. 4D and 5A). In these cases, it is clear that SLVI had undergone more extensive alteration of base pairing and structure. Previous studies of SLVI, independent of LRA disruption, were shown to result in alteration to the translation phenotype (Vassilaki et al., 2008). Without a greater understanding of the RNA structure in this region, for example, by expanding the region analysed by SHAPE mapping, we do not think a complete interpretation of the relationship between RNA structure and phenotype is possible. In addition, the HCV IRES is known to be flexible and dynamic (Pérard et al., 2013) and it is possible that the averaged data obtained from SHAPE mapping may obscure some of this flexibility. Indeed, the error bars covering some data points may suggest a much more dynamic situation at the single molecule level than is represented by the overall RNA population while in solution.

Notwithstanding the potential for the LRA forming during HCV infection in vivo, we went on to analyse HCV translation and replication from templates in which we engineered the 'open' or 'closed' conformations by mutagenesis of the complementary pairing critical for LRA formation. We reasoned that, if dramatic differences were observable, this might indicate a role for the LRA and our ability to detect the 'closed'
structure in vitro might not be representative of conditions in vivo. For example, cellular proteins could influence the formation of the LRA or miR122 may be limiting or absent at certain stages of the replication cycle.

We investigated translation from a core-extended template transfected into Huh 7.5 cells. As expected from previous studies (*Kim et al., 2003*), extension of core-encoding sequences—to encompass SLVI—reduced reporter gene expression by ∼50%. When normalised to this lower level of translation, the template engineered to adopt the 'closed' conformation (JFH1-CEtrans-S1/L) exhibited levels of translation essentially indistinguishable from the control. In contrast, the template unable to recruit miR122 to the S1 seed site (JFH1-CEtrans-S1) and therefore solely adopting the 'open' conformation, exhibited a slight but significant increase in translation (Fig. 6B). Compared to its vital role in viral replication, the role of miR122 in stimulating HCV translation is not as clear, with some reports suggesting mutations in miR122 seed sites do not lead to changes in translation (*Jopling et al., 2005*) and others observing decreases in translation upon mutation of either, or both, seed sites (*Jangra, Yi & Lemon, 2010*). Our results for JFH1-CEtrans-S1 and JFH1-CEtrans-S1/L/R, both adopting an 'open' conformation, would support the studies by *Jopling et al. (2005)*. Despite inhibition of miR122 binding at the S1 site, both templates were still able to recruit miR122 to the S2 site (Fig. 2), thus maintaining parental, or near-parental levels of translation. Therefore, templates of known conformation with regard to the LRA did not show a correlation between a closed structure and a reduction in translation, and opening the structure only led to a minor increase in translation in one template, JFH1-CEtrans-S1. In addition, relative translation phenotypes in HeLa cells were comparable to Huh 7.5 cells demonstrating that the presence of SLVI is sufficient to account for the reduction in translation observed between JFH1-luc-trans:ΔNS5B and JFH1-CEtrans (Figs. 6B and 6C). The presence of SLVI has also been shown to influence translation through modulation of RNA interactions involving domain 5BSL3.2 (*Ventura et al., 2017*). Such interactions may contribute to the translation phenotypes we have demonstrated here to be independent of the LRA.

In contrast to the results obtained using translation templates bearing mutations to destroy/recreate the LRA, analysis of genome replication bearing identical mutations was easier to interpret. In these, any mutation of the S1 miR122 seed site reduced replication by ∼2 $\log_{10}$ at 28 h post-transfection (Fig. 6C) and this phenotype could be readily and fully rescued by provision of a complementary S1-miR122 in trans. These results imply that the structures adopted by sequences predicted to be involved in the LRA either have no influence on genome replication, or that the influence is negligible when compared to the known impact of reduced miR122 binding (*Jopling, Schütz & Sarnow, 2008*; Jopling et al.,2005; *Li et al., 2013*).

Taken together, these studies suggest that the LRA between the S1 binding site and the basal stem of SLVI is unlikely to contribute to temporal control of genome translation and replication. In the absence of introduced complementary mutations we could only demonstrate formation of the LRA-associated 'closed' structure under very specific conditions in which miR122 was absent (Fig. 3B). Whether such conditions occur

in vivo is unclear. An estimated 66,000 copies of miR122 have been reported in hepatocytes (*Jopling, 2012*) and *Luna et al. (2015)* have reported that HCV replication de-represses cellular targets of miR122, implying that the replicating virus genome acts as a 'sponge' to sequester miR122. Although the latter suggests that miR122 is limiting, it is unlikely to be early in the infection cycle. At this time a small number of genomes are present and temporal control of genome translation and replication is likely critical outside the compartmentalisation offered by membrane-bound replication complexes (*Wölk et al., 2008*; *Miyanari et al., 2003*).

## CONCLUSIONS

Conformational changes in RNA structure are one method by which RNA viruses can modulate essential genome functions such as translation and replication. In this study, we demonstrate that one such conformational change, the LRA, involving complementary sequences in the HCV IRES and a core gene stem-loop structure, is unlikely to act as a modulator between translation and replication. We have shown that switching between the 'open' and 'closed' conformations is a miR122 dependent process and confirmed that presence of the core stem-loop structure SLVI results in a drop in translation activity. However we have demonstrated that templates preferentially forming either the 'open' or 'closed' conformation are not associated with any translation or replication phenotypes. Instead we propose that the stem-loop structure SLVI, mediates translation via other, as yet undefined mechanisms.

### Funding

This work was supported by the UK Medical Research Council (G1100139). The funders had no role in study design, data collection and analysis, decision to publish, or preparation of the manuscript.

### Grant Disclosures

The following grant information was disclosed by the authors:
UK Medical Research Council: G1100139.

### Competing Interests

The authors declare that they have no competing interests.

### Author Contributions

- Kirsten Bentley conceived and designed the experiments, performed the experiments, analysed the data, contributed reagents/materials/analysis tools, prepared figures and/or tables, authored or reviewed drafts of the paper, approved the final draft.
- Jonathan P. Cook conceived and designed the experiments, performed the experiments, contributed reagents/materials/analysis tools, approved the final draft.
- Andrew K. Tuplin conceived and designed the experiments, performed the experiments, contributed reagents/materials/analysis tools, approved the final draft.

- David J. Evans conceived and designed the experiments, contributed reagents/materials/ analysis tools, authored or reviewed drafts of the paper, approved the final draft.

## Data Availability

The raw data are provided in the Supplemental Files.

## Supplemental Information

Supplemental information for this article can be found online at http://dx.doi.org/10.7717/ peerj.5870#supplemental-information.

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
