# Peer review of "Structural and functional analysis of the roles of the HCV 5′ NCR miR122-dependent long-range association and SLVI in genome translation and replication"

_PeerJ, doi:10.7717/peerj.5870_

## Round 0.1 · original submission · Major Revisions

I personally like your paper. However, given that negative results could arise for a variety of reasons and that RNA structures can potentially take many more states than "open" and "close", I am inclined to agree with Reviewer #3 that significant revision is required. All suggested changes from reviewers are relatively specific, and I look forward to see your revised manuscript together with a point-by-point response to comments from reviewers.

Reviewer 1 ·

Basic reporting

This manuscript analyses the “open and closed conformation” of the 5´UTR of HCV viral RNA extended up to nt 174 of core, hence including domain VI. First, by using SHAPE methodology, the authors confirm the existence of reactive/unreactive nt within domain VI (suggesting unpaired/paired nt, respectively). Then, the wt RNA, and several mutations disrupting and restoring the hairpin of domain VI, as well as the target sequences of miR122 at the 5´end, are used to explore the role of the closed and open conformation of the 5´UTR in translation and replication of HCV RNA in HuH7 cells (miR122 positive) or HeLa cells (miR122 negative). The authors conclude that domain VI is involved in the formation of the open and closed form of the HCV IRES, as previously reported. However, no effect was observed on translation or replication in the presence or absence of miR122. They conclude that the involvement of domain VI in translation control should be related to a still unknown event, different than formation of closed and open conformation. The manuscript is well written, despite the results are not increasing our understanding of the involvement of the “open and closed conformation” of the HCV IRES region in the biology of HCV RNA.

Experimental design

The authors present a simple scenario concerning two mutually exclusive HCV RNA structures predicted to adopt open and closed conformations (l. 104-108), which involve two miR122 binding sites at the 5´end, and at the 3´end, the base of stem-loop VI. However, it has been previously shown that the HCV 5´UTR encompassing the IRES element adopts flexible, dynamic conformations in solution (i. e., Perard et al). Comments to a flexible nature of this region regarding the interpretation of the results obtained by SHAPE are lacking.
According to the results stated in l. 307-311, the authors could not map the accessibility to S1 miR122 binding site (nt 1-20). Therefore, the conclusions regarding the open and close conformation are not valid, as only the accessibility to SLVI is accurately determined. This comment also applies to the sentence in l. 348-349 (… despite the inability to measure the reactivity of nt within the S1 site). Moreover, figures 3 to 5 show the changes in reactivity pertaining nt 427-507. The authors refer to same NMIA reactivity and structural predictions as data not shown for a variety of unmodified templates. As it stands, this sentence is not informative. The authors should show the SHAPE reactivity of the entire 5´UTR.

Validity of the findings

see above

Additional comments

see above

Reviewer 2 ·

Basic reporting

The work presented by Bentley and co-workers, represents an interesting structural and functional study of the long range association between domain SLVI and the S1 miR122 binding site within the 5’ end of the hepatitis C RNA genome. They confirm the existence of such interaction but they question its functionality. However they point out a functional role of domain SLVI as regulator of the viral translation by a still undefined mechanism. This is a very interesting observation which is worthy to pursue.
The manuscript is well written, clear and concise. The references provided are appropriate in general. However this reviewer suggests adding in line 76 (introduction) more recent reviews covering the cited topic, if any, and in particular would be interesting for readers to see references of recent reviews focusing in the functional role of RNA structural domains in the HCV genome.
1. A formatting correction, a space should be added in between the number and the units all throughout the text (e.g., 4 h; 5 nM; and so on)
2. The word “conformation” is misspelled several times, instead “confirmation” is written. Authors might want to carefully revise all the manuscript.

Experimental design

The work is well-defined and executed, and the structural analysis is clear and convincing. The experimental work is described is sufficient detail.
It would be helpful for the readers that the authors add in figure 1B a schematic representation of the pJFH1-luc-trans:ΔNS5B, which has been previously described by the authors (Tuplin et al., 2015). Although they provide the reference it would help to see it in comparison with the other constructs used in this work.

Validity of the findings

The conclusions are well sustained by the provided results; however there are a couple of issues that are worthy to be addressed.
1. Results shown in figure 3, clearly suggest that in the absence of miR122 the LRA between S1 site and the 5’ end of the SLVI domain takes place leading to the formation of the closed conformation as it is interpreted by the authors (lines 343-348). However these results are not in good agreement with the data shown in the following section as it is stated in its heading (line 359) saying that the LRA is favored ONLY when … SLVI structure is disrupted.
Authors should provide an explanation of this apparent discrepancy.
2. The results are based on the use of sequence mutants that induce the desired structural changes in the base of the SLIV stem and/or the lack of binding at the S1 site. The phenotypic analysis of these mutants (effect on viral translation and replication) allow the authors to draw the conclusion of the lack of functionality of the switch between the “open” and “closed” conformation and further to propose a non-described translation regulation role for the SLVI domain. However, the authors have not considered the potential pleiotropic effect of structural rearrangement induced by mutants that may hide the functional role of LRA.
In addition they have not discussed the potential phenotypic role of the S2 miR122 binding site in the context of a mutant S1 site.

Reviewer 3 ·

Basic reporting

I recomned the authors to improve significativelly the correspondence of their findings to already published work and their claims to be limited to their results. They have to introduce the limitations of the methodology employed in their work, I provide some lines. And absolutely not, they can not, contradict previous work with a -data not shown-.
For the rest it is OK.

Experimental design

The authors bases their experimentation in a quite new technique, which has the nice name of SHAPE. This is being widely used. But is it has some shortcomings, and specially for the problem the authors are trying to solve.

- In difference for the original description fo the strucutre of stem-loop VI, whcih was performed accordign well stablished standarts, using RNAse V1 and ssRNAses, here the authors have swicht to a methodolgy of much lower euristic value:
- it is an indirect technique: what is read is the blocking of a primer extension of a cDNA reaction, and not the RNA itself.
- it suffers of polarity problem, the strucuture can only be mapped from 3' end.
- it has no way of falsability, in contrast to RNAses A, T1 their reactivities should not coincide with RNase V1 and viceversa, and if this happens, then there is an interesting situation to analyse.
- If two or more structures are present at the same moment these will be promediated at each base, so there is no way to ascertain if this is the case.

Validity of the findings

The results are fine, the mutational analysis greatly support previous biochemical analysis by Diaz-Toledano. it is a high complementary analysis. The authors should not try to ignore the work of Diaz-Toledano. The authors should distinghuish what is a computer prediction, from what has been already probed by several bicohemical methodologies, and also distinguish the history of findings. The first long range annealing propossed in HCV is that of 5' and stem loop VI. This was deduced by two groups in parallel, Honda&Lemon and Kim et al, but no one of this groups did structural analisis , they deudced the structure from functional or phylogentic analysis, so their findings are the first but indirect respect to the structure. The first structure determination of the LRA is that of Begurisitain et al, by direct detection of the LRA duplex, who also showed at least 80% of the form closed in respect to the opened.
Then the first in demosntrate opening by miR122 is Diaz-Toledno et al. Some of the prhases of authors that say that this structure has been proposed, they should say probed. Please review all the paper.
I do not understand why the results in vivo may not reflect the in vitro LRA formation.

Additional comments

-There is a substantial contradiction between the expectations from the title and the end of the abstract. -Despite the authors do not find correlation between Open/Closed forms, with translation or replication, this can not be extended to the global viral phenotype. Leibniz the German matematician said: when we negate something we are wrong. I wil try to rethink the results and avoid the negative proposal.
- The second big parapraph of the introduction, is too large and it is not quite relevant for the paper, this are other Long range interactions, mostly out of the focus.

---

## Round 0.2 · accepted · Accept

I enjoy reading your article. Hope that you will work towards a more quantitative modelling of the regulatory process. Keep up your good work. Congratulations.

Best.
Xuhua
http://dambe.bio.uottawa.ca

# Reviewer 3 ·

Basic reporting

OK

Experimental design

OK

Validity of the findings

OK

Additional comments

OK